# OASIS: ILP-Guided Synthesis of Loop Invariants

**Sahil Bhatia**[†], **Saswat Padhi**[§], **Nagarajan Natarajan**[†], **Rahul Sharma**[†], **Prateek Jain**[†]

[†]Microsoft Research India

[§]University of California, Los Angeles

{t-sab,nagarajn,rahsha,prajain}@microsoft.com, padhi@cs.ucla.edu

## Abstract

Automated synthesis of inductive invariants is an important problem in software verification. We propose a novel technique that is able to solve complex loop invariant synthesis problems involving large number of variables. We reduce the problem of synthesizing invariants to a set of integer linear programming (ILP) problems. We instantiate our techniques in the tool OASIS that outperforms state-of-the-art systems on benchmarks from the invariant synthesis track of the Syntax Guided Synthesis competition.

## 1 Introduction

Program verification aims to provide a strong guarantee on the correctness of an implementation by formally proving that it meets a desired property, such as termination, correctness of assertions, memory safety, and more. Inferring inductive invariants is one of the core problems of program verification. Many verified-programming environments [6, 8, 14] require users to furnish the right invariants. However, it can be quite challenging, even for expert programmers, to annotate these invariants for simple practical cases.

Loop invariant is a predicate over the program state that is preserved across each iteration of the loop. Consider a simple loop, **while** $G$ **do** $S$, which executes the statement $S$ until the condition (loop guard) $G$ holds and then it halts. Then, a predicate $\mathcal{I}$ is said to be a sufficient loop invariant and validates a Hoare triple $\{\rho\}$ **while** $G$ **do** $S$ $\{\phi\}$, if it satisfies the following three verification conditions:

$\text{VC}_{\text{pre}}:$    $\rho \implies \mathcal{I}$, i.e., $\mathcal{I}$ must hold immediately before the loop

$\text{VC}_{\text{ind}}:$    $\{G \wedge \mathcal{I}\}$ $S$ $\{\mathcal{I}\}$, i.e., $\mathcal{I}$ must be inductive (hold after each iteration)

$\text{VC}_{\text{pos}}:$    $\neg G \wedge \mathcal{I} \implies \phi$, i.e., $\mathcal{I}$ must certify the postcondition upon exiting the loop

Traditionally successful approaches are based on enumerating all possible expressions for the invariant. Recent prior work [13, 15] propose using machine learning (ML) and continuous optimization techniques to synthesize invariants. A key issue is that while continuous optimization is highly efficient for solving a problem approximately, invariant synthesis demands finding an *exact* solution.

We propose a tool OASIS[1] that takes as input logic formulas which encode the verification of safety properties of programs over integer variables and outputs inductive invariants that are sufficient to prove the properties. To this end, OASIS employs new ML algorithms for the well-known *binary classification* problem: the learner's goal is to find a *classifier* that *separates* positive and negative examples. In the context of invariant synthesis, an *example* is a program *state* that maps variables to integers. OASIS makes the following contributions.

First, OASIS uses binary classification to infer relevant and irrelevant variables (Section 2.2). It uses symbolic execution to generate *reachable* states (positive examples) and *bad* states (negative examples), which are backward reachable from states that violate the safety properties. Then it finds a *sparse* classifier and we classify the variables occurring in the classifier as relevant. If a variable is

---

[1]    The name OASIS stands for **O**ptimization **A**nd **S**earch for **I**nvariant **S**ynthesis.

absent from the classifier and it is possible to separate samples of reachable states from bad states without using the variable then it is likely to be irrelevant to the invariant.

Second, OASIS uses a learner to synthesize Boolean features from data. OASIS is based on LOOP-INVGEN [10, 11] that breaks down the problem of invariant synthesis into many small binary classification tasks and uses Escher [3] to find *features* that solve them. Specifically, Escher exhaustively enumerates all features in increasing size till it finds one that separates the positive examples from the negative examples in the small task. OASIS replaces Escher with a learner to find such features. OASIS uses the same learner to solve both these problems, i.e., inferring relevant variables and inferring features.

We evaluate OASIS on 403 benchmarks from the invariant (Inv) track of the Syntax Guided Synthesis (SyGuS) competion held in 2019 [2]. Our evaluation shows that OASIS significantly improves invariant synthesis on these benchmarks.

## 2 Overview of Our Approach

### 2.1 Relevant Variables

---

**Algorithm 1** OASIS framework for scaling loop invariant inference

---

1    **func** OASIS($\langle$PRE, TRANS, POST$\rangle$ : Verification Problem, $\vec{\sigma}_+$ : States, $\vec{\sigma}_-$ : States)
2       Classifier $\mathcal{C} \leftarrow$ LEARN($\vec{\sigma}_+, \vec{\sigma}_-$)
3       **if** $\mathcal{C} = \bot$ **then return** $\bot$
4       Variables $\vec{r} \leftarrow$ FILTERVARIABLES($\mathcal{C}$)
5       **do parallel**
6          **in thread** 1 **do**
7             $\vec{\sigma} \leftarrow$ FINDPOSCOUNTEREXAMPLE($\langle$PRE, TRANS, POST$\rangle, \mathcal{C}$)
8             **if** $\vec{\sigma} \neq \bot$ **then return** OASIS($\langle$PRE, TRANS, POST$\rangle, \vec{\sigma}_+ \cup \{\vec{\sigma}\}, \vec{\sigma}_-$)
9          **in thread** 2 **do**
10            $\vec{\sigma} \leftarrow$ FINDNEGCOUNTEREXAMPLE($\langle$PRE, TRANS, POST$\rangle, \mathcal{C}$)
11            **if** $\vec{\sigma} \neq \bot$ **then return** OASIS($\langle$PRE, TRANS, POST$\rangle, \vec{\sigma}_+, \vec{\sigma}_- \cup \{\vec{\sigma}\}$)
12          **in thread** 3 **do**
13            $\mathcal{I} \leftarrow$ RELINFER($\langle$PRE, TRANS, POST$\rangle, \vec{\sigma}_+, \vec{\sigma}_-, \vec{r}$)$\big|_{\text{timeout} = \tau}$
14            **if** $\mathcal{I} \neq \bot$ **then return** $\mathcal{I}$

---

Our core framework, called OASIS, is outlined in Algorithm 1. OASIS accepts a verification problem (encoded as a triple $\langle$PRE, TRANS, POST$\rangle$), and some sampled positive ($\vec{\sigma}_+$) and negative ($\vec{\sigma}_-$) program states typically sampled randomly. We first invoke the LEARN function with these sampled states to learn a predicate $\mathcal{C}$ that separates $\vec{\sigma}_+$ and $\vec{\sigma}_-$ We detail the LEARN function in Section 3, which utilizes machine-learning techniques to efficiently find a sparse separator for $\vec{\sigma}_+$ and $\vec{\sigma}_-$. In line 4 we drop irrelevant variables, those that do not affect the prediction of the classifier over $\vec{\sigma}_+ \cup \vec{\sigma}_-$, and consider the remaining variables $\vec{r} \subseteq \vec{x}$ ($\vec{x}$ is the set of all variables in the program) to be a candidate set of relevant variables.

After a set $\vec{r}$ of relevant variables is identified, in lines 2 – 11, we try to refine the set of relevant variables and find a sufficient invariant over them in parallel. In particular, we execute the following three threads in parallel:

1. one that attempts to find a *positive* state misclassified by the classifier
2. one that attempts to find a *negative* state misclassified by the classifier
3. one that runs invariant inference using the currently identified relevant variables

### 2.2 Refining Relevant Variables

We now detail our procedures for refining a set of relevant variables. Each of these procedures returns a program state that is misclassified by the current classifier $\mathcal{C}$, which is then used to learn a new classifier, and thus a new set of relevant variables.

The FINDPOSCOUNTEREXAMPLE procedure identifies *positive* misclassifications — a reachable program state $\vec{\sigma}$ that the classifier labels as a negative state, i.e., $\neg\mathcal{C}(\vec{\sigma})$. To identify such states,

we gradually expand the frontier of reachable states starting from the precondition PRE and then repeatedly applying the transition relation TRANS.

The FINDNEGCOUNTEREXAMPLE procedure works in a very similar manner and identifies *negative* misclassifications — a bad program state $\vec{\sigma}$ (one that would lead to violation of the final assertion) that the classifier labels as a positive state, i.e., $\mathcal{C}(\vec{\sigma})$. To identify such states, we gradually expand the frontier of known bad states starting from those that violate the postcondition POST and then repeatedly reversing the transition relation TRANS.

Once we have a set of relevant variables from the learned classifier, we run our invariant inference algorithm RELINFER (in thread 3) with these variables together with all the positive states ($\vec{\sigma}_+$) and negative states ($\vec{\sigma}_-$) sampled so far. OASIS builds over LOOPINVGEN, we refer to the LOOPINVGEN paper [10] for details on the RELINFER algorithm. They key difference is while LOOPINVGEN learns features over all variables $\vec{x}$ in the program, RELINFER only learns features over $\vec{r}$.

## 3 The ILP Formulation

In this section, we formulate the problem of generating a classifier that separates positive program states from negative program states. By default, the output classifier predicate can use any of the program variables. If we restrict the classifier to use only a subset $\vec{r}$ of variables then we first project the examples to $\vec{r}$ and then learn a classifier over the projected states. Let $x$ denote a vector of program variables that can occur in the classifier. We model the problem of inferring a classifier $h : \mathbb{Z}^{|x|} \to \{\mathsf{True}, \mathsf{False}\}$ as a search problem over the following class of CNF predicates with $nc$ denoting the number of conjuncts and $nd$ the number of disjuncts in each conjunct:

$$\mathcal{H}_{\mathrm{CNF}} = \left\{ \bigwedge_{c\in[C]} \bigvee_{d\in[D]} \langle \mathbf{w}_{cd}, x \rangle + b_{cd} > 0 \right\}. \tag{1}$$

where $b \in \mathbb{Z}$ and $\langle w, x \rangle + b$ is an inner product between a vector $w \in \mathbb{Z}^{|x|}$ and $x$. We use $[n]$ to denote the list $\{0, 1, \ldots, n-1\}$.

Consider the search problem (1) above: formally, we want to find a predicate $h \in \mathcal{H}_{\mathrm{CNF}}$ that accurately classifies a given set of labeled program states $\{\sigma_n, y_n\}_{n=1}^N$, where $y_n \in \{0, 1\}$. It is convenient to think of $h$ as a tree of depth 3: the program variables form the input layer to the linear inequalities, which are grouped by $\bigvee$ operators to yield disjunctive predicates. The root node is the $\bigwedge$ operator that represents conjunction of the predicates represented by the second layer. The reduction of the search problem to ILP is given as follows.

(**Input layer: Thresholded polynomials**) Write $z_{ncd} = \mathbb{1}\left\{ \langle \mathbf{w}_{cd}, \sigma_n \rangle + b_{cd} > 0 \right\} \in \{0, 1\}$. This is captured by the following constraints, for a sufficiently large integer $M$:

$$\forall n \in [N], c \in [C], d \in [D], \quad -M(1 - z_{ncd}) \;<\; \langle \mathbf{w}_{cd}, \sigma_n \rangle + b_{cd} \;\leq\; M z_{ncd},$$
$$\mathbf{w}_{cd} \in \mathbb{Z}^{|x|}, b_{cd} \in \mathbb{Z}, z_{ncd} \;\in\; \{0, 1\}. \tag{2}$$

(**Middle layer: Disjunctions**) Note that the value of the $c$-th conjunct on a given input $\sigma_n$ corresponds to summing $z_{ncd}$ over $d$, i.e., write $y_{nc}^\vee = \bigvee_{d\in[D]} z_{ncd}$. This is captured by:

$$\forall n \in [N], c \in [C], \quad -M(1 - y_{nc}^\vee) \;<\; \sum_{d\in[D]} z_{ncd} \;\leq\; M y_{nc}^\vee,$$
$$y_{nc}^\vee \;\in\; \{0, 1\}. \tag{3}$$

(**Final layer: Conjunction**) The predicted label on a given input state is given by a conjunction of the above disjunctions. Requiring that the predicted label match the observed label for each example is equivalent to the following constraints:

$$\text{for } n \in [N] \text{ s.t. } y_n = 1, \sum_{c\in[C]} y_{nc}^\vee \;\geq\; C\,,$$
$$\text{for } n \in [N] \text{ s.t. } y_n = 0, \sum_{c\in[C]} y_{nc}^\vee \;\leq\; C - 1\,. \tag{4}$$

The search problem can now be stated as the ILP problem: *find a feasible integral solution* $\{z, y^\vee, \mathbf{w}, b\}$ *subject to the constraints Equations (2), (3) and (4) combined.*

Now, consider the problem of learning *generalizable* predicates (2). To this end, we follow the Occam's razor principle – seeking predicates that are "simple" and have been shown to generalize better [10]. Simplicity in our case can be characterized by the size of the predicate clauses and the magnitude of the coefficients. One way to achieve this is by constraining the $L_1$-norm of the coefficients $\mathbf{w}$. Note that $L_1$-norm can be expressed using linear constraints: $\|\mathbf{w}\|_1 = \langle \mathbf{1}, \mathbf{w}^+ + \mathbf{w}^- \rangle$, where $\mathbf{w}^+ \geq 0$ and $\mathbf{w}^- \geq 0$ (componentwise inequality) such that $\mathbf{w} = \mathbf{w}^+ - \mathbf{w}^-$.

To learn shortest predicate we penalize the *inclusion* of variables in the solution by using a penalty $\mu$ where $\mu_j = 0$ iff $\forall c \in [C].\forall d \in [D]. (\mathbf{w}_{cd})_j = 0$. Our final objective function combines both the penalties:

$$\min_{\mathbf{w}, \mathbf{w}^+, \mathbf{w}^-, b, z, y^\vee, \mu} \sum_{c \in [C], d \in [D]} \langle \mathbf{1}, \mathbf{w}_{cd}^+ + \mathbf{w}_{cd}^- \rangle \;\; + \;\; \lambda \langle \mathbf{1}, \mu \rangle$$

subject to Equations (2), (3) and (4), and

$$\mathbf{1} - M(\mathbf{1} - \mu) \;\; \leq \;\; \sum_{c \in [C], d \in [D]} \mathbf{w}_{cd}^+ + \mathbf{w}_{cd}^- \;\; \leq \;\; M\mu,$$

$$\forall c \in [C], d \in [D], \quad \mathbf{w}_{cd} \;\; = \;\; \mathbf{w}_{cd}^+ - \mathbf{w}_{cd}^-,$$

$$\mathbf{w}_{cd}^+ \geq 0, \;\; \mathbf{w}_{cd}^- \geq 0, \;\; \mu \;\; \in \;\; \{0,1\}^{|x|}. \tag{5}$$

## 4  Experimental Evaluation

We have implemented OASIS using the LOOPINVGEN [10] framework in OCaml, and using Z3 [7] as the theorem prover for checking validity of the verification conditions. We implemented our logic for reducing the classification problem to ILPs in a Python script, which discharges the ILP subproblems to the OR-Tools [1] optimization package from Google. We evaluate OASIS on commodity hardware — CPU-only machines with up to 32 GB RAM running Ubuntu Linux 18.04.

**Solvers and Benchmarks.** We compare OASIS, against three tools: (1) LOOPINVGEN [10] which uses data-driven invariant synthesis (2) CVC4 [5, 12] which uses a refutation-based approach (3) DRYADSYNTH [9] which uses a combination of enumerative and deductive synthesis (cooperative synthesis). CVC4 and LOOPINVGEN are respectively the winners of the invariant-synthesis (Inv) track of SyGuS-Comp'19 [2] and SyGuS-Comp'18 [4]. Recently, [9] showed that DRYADSYNTH's cooperative synthesis technique is able to perform better than LOOPINVGEN and CVC4 on invariant synthesis tasks. We evaluate our technique on 403 instances which were part of the SyGuS-Comp'19 [2] and were the invariant synthesis benchmarks used to evaluate [9]. All these instances require quantifier free reasoning over linear arithmetic.

| Tool | Solved (out of 403) |
|---|---|
| CVC4 | 287 |
| DRYADSYNTH | 346 |
| LOOPINVGEN | 272 |
| OASIS | **353** |

Table 1: Comparison between OASIS and SyGuS competitors, on the 403 instances which were part of the SyGuS-Comp'19 [2].

**Results.** We report the number of instances each tool solves with a timeout of 30 minutes[2] in Table 1. OASIS synthesizes sufficient loop invariants on **353** instances, **7** more than the second best tool and **66** more than CVC4, the winner of invariant-synthesis (Inv) track of SyGuS-Comp'19. OASIS is able to solve **13** instances which no other tool can solve.

---

2  [9] uses a timeout of 30 minutes and we keep the same timeout.

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
