# OpenReview forum: "Oasis: ILP-Guided Synthesis of Loop Invariants"
_NeurIPS.cc/2020/Workshop/CAP — NeurIPS 2020 CAP Workshop_

### Official Review · AnonReviewer1 · 2020-10-30
**OASIS**

**Rating:** 7
**Confidence:** 3

**Review:**

This paper presents a technique for solving loop invariant synthesis problems. The technique, called OASIS, learns to identify and refine a set of relevant variables for use within an invariant inference algorithm. This allows an invariant inference algorithm to only learn features over the relevant variables, reducing the difficulty of the search problem.

The model is tested on SyGuS invariant synthesis problems, and experimental results show that the model outperforms strong baselines.

Overall, this work seems sound and relevant, and would make a good addition to the workshop.

---

### Decision · Program_Chairs · 2020-11-02

**Decision:**

Accept

**Comment:**

As the review is positive, I recommend acceptance.